# Factors and mediators impacting the number of undergraduate research mentees at a research-intensive Hispanic-serving institution

Angelica Monarrez[1☉], Lourdes Echegoyen[2], Danielle Morales[3], Maria Aleida Ramirez[4], Diego Seira[5], Amy Wagler[6☉]*

1 Research Evaluation and Assessment Services, The University of Texas at El Paso, El Paso, Texas, United States of America, 2 Campus Office of Undergraduate Research Initiatives, The University of Texas at El Paso, El Paso, Texas, United States of America, 3 Department of Sociology, Worcester State University, Worcester, Massachusetts, United States of America, 4 Department of Mathematical Sciences, The University of Texas at El Paso, El Paso, Texas, United States of America, 5 Office of Data Science, NIAID, NIH, Bethesda, Maryland, United States of America, 6 Department of Public Health Sciences, The University of Texas at El Paso, El Paso, Texas, United States of America

☉ These authors contributed equally to this work.
* awagler2@utep.edu

**Data Availability Statement:** Data cannot be shared publicly because of issues of confidentiality. The data consists of faculty information that is not public and due to some small sample sizes for

## Abstract

Engaging in undergraduate research experiences is known to have broad and positive impacts on college students. Despite the benefits, achieving faculty buy-in and support can be challenging even when faculty have strong research funding. In order to understand how to better support undergraduate research programs, we applied quantitative models to explore how the impact of research funding is mediated by faculty beliefs about undergraduate research mentoring in STEM disciplines. The results indicate that faculty characteristics and beliefs about the benefits and barriers of mentoring undergraduate students in research impact the number of students mentored even when accounting for research funding levels of the individual faculty. Practical recommendations are presented based on the models from this research project in order to provide insight into how centers or units on a campus may work with faculty to support and encourage mentoring of undergraduates in research in the biomedical sciences.

## Introduction

Undergraduate research experiences (UREs) are increasingly valued as high impact experiential educational activities that have broad and long-standing positive impacts on college students [1, 2]. Participation in UREs, whether situated in a structured program or not, are known to increase students' research self-efficacy, science identity and impact graduate school enrollment [3–9]. Additionally, faculty members who have been part of a URE are motivated to engage with undergraduate students and highly value these mentoring opportunities for a

certain faculty characteristics, the faculty could reasonably be identified if the data were made public. If data is requested, please email Dr. Rafael Aguilera, Research, Evaluation, and Assessment Center, UTEP (raguilera3@utep.edu).

**Funding:** -LE,AW -Research reported in this paper was supported by the National Institute of General Medical Sciences of the National Institutes of Health under linked Award Numbers RL5GM118969, TL4GM118971, and UL1GM118970. The content is solely the responsibility of the authors and does not necessarily represent the official views of the National Institutes of Health. -https://www.nih.gov/ -The funder played no role in study design, data collection or analysis, decision to publish, or preparation of the manuscript.

**Competing interests:** The authors have declared that no competing interests exist.

variety of professional and personal reasons [10]. Though the effectiveness of URE programs is often well understood by faculty members, not all prioritize mentoring undergraduate students in research. This is not necessarily due to a lack of interest in mentoring undergraduates. In fact, there are a variety of factors that may influence whether or not a faculty member includes undergraduate students in their research endeavors [10]. This study probes these factors to better understand how they affect the number of undergraduate research mentees working with an individual faculty member, while controlling for the level of research funding of the faculty surveyed.

## Literature review

Faculty mentorship is one of the central elements of UREs, and the myriad benefits for undergraduate mentees include improved critical thinking, increased academic achievement and retention, persistence to STEM degree completion, clarification of career plans, and improved preparedness or desire for graduate study [11]. Particularly for underrepresented minority students, faculty mentors could effectively arouse, promote, and sustain their STEM career interests [12]. However, the positive outcomes of faculty mentorship in UREs are not universal. Numerous studies have examined the conditions under which mentoring is most effective and identified factors influencing mentor-mentee relationships [13–18].

Most empirical research examining mentoring relationships has focused on mentees' perspectives. However, it is also important to understand mentorship dynamics from the mentor's perspective, because faculty mentors play a key role in UREs. They directly influence students' learning outcomes, their decisions to attend graduate school, and their career choices [19]. Hence, committed mentors are crucial to the success of the mentees [20, 21]. Among limited studies, Eagan [21] analyzed data from 4,832 STEM faculty across 194 institutions and found that faculty who worked in the life sciences, such as biomedical sciences, and those who received government funding for their research were more likely to involve undergraduates in research projects. Faculty at historically Black colleges were significantly more likely to involve undergraduate students in research than their colleagues at predominantly White institutions and Hispanic-serving institutions [22]. Webber [23] examined approximately 40,000 responses to the Faculty Survey of Student Engagement at over 450 four-year institutions. Their findings revealed that both individual and institutional characteristics predicted faculty participation in UREs. At the individual level, they found that African American faculty and faculty with doctorate degrees were more likely to participate in UREs than their colleagues. Junior faculty, male faculty, faculty with larger course loads, and faculty with more teaching experience were slightly more likely to participate in URE than their colleagues. Other researchers pointed out that there are particular obstacles for faculty to become an URE mentor. For example, Johnson [24] suggested that many colleges and universities only reward faculty for funded research and publications, but not teaching and mentoring. Chopin [25] argues that some faculty members hesitate to become involved in UREs because they believe that it is time-consuming, and undergraduate students need more training, explanation, and supervision than graduate students. On the other hand, some faculty members are willing to involve undergraduates in their research because they believe those students can receive significant educational benefits from the research experience and also when they believe the institution values undergraduate research mentoring [19, 26, 27].

Although the motivations for and benefits of mentoring undergraduate students have been recognized and appreciated [10, 28] and others have identified factors leading to undergraduate mentoring [27], the factors influencing how many students are mentored is not well understood, particularly when controlling for factors already known to affect undergraduate

mentoring. There are clear and salient factors that are likely to influence how many students overall are mentored in a biomedical research lab. For example, the amount of funding available to the lab, the area or focus or discipline of the lab, the size of the lab, the success of the faculty's research endeavors, as well as common and persistent obligations in teaching and service that take faculty time away from research and mentoring. This paper aims to develop a theoretical model to understand faculty mentors' attitudes associated with the number of undergraduate students they mentor, and test the model using empirical data to identify factors that enable or constrain faculty engagement in undergraduate research mentoring.

## Theoretical framework

We theorize that social exchange theory (SET) [29, 30] is the framework by which faculty members involved in URE determine how many students they mentor. There are core assumptions of the SET framework that can explain the relative prioritization of research mentoring among other research, teaching, and service activities for otherwise similar faculty members. Moreover, this framework can guide the interpretation of the modeling results. One of the core assumptions of SET is that people *chase rewards and avoid punishment*. For example, if an activity brings about a sense of contentment, people will seek these out rather than avoid them. A second core assumption of the SET framework is that people attempt to obtain *maximum benefit while expending minimum cost*. For example, if a person can obtain the same benefit from two activities and one requires far less time, most will tend to seek out the activity with less time commitment. And, the third SET assumption, is that when faced with a choice, people will *weigh the potential benefit and cost and decide according to these perceptions*. For example, if a person has a choice about participating in an activity that brings a sense of satisfaction, they will participate if the sense of satisfaction is worth the cost. Exercise is a good example of this since it requires effort and time, but many judge the benefits to be worth the cost. Here, we take the perspective of Emerson [30], who stated that resource availability, power dynamics, and dependencies involved therein are the primary factors involved. Emerson [30] believed that social relationships vary by setting, purpose and the kind and extent of resources being exchanged. This makes sense in our context of URE mentoring since there are well-identified benefits and barriers of engaging in mentoring [10].

Though the identified benefits and barriers of UREs mentoring are the primary research outcome of the Monarrez et al. [10] study, we note that several factors were identified in that study that affect the presence of specific benefits and barriers held by faculty. In particular, the gender, discipline, and rank of the biomedical research faculty members were found to affect specific beliefs about undergraduate research mentoring. Moreover, institutional characteristics, such as valuing undergraduate research were also found to affect these beliefs. These factors were found to inform the beliefs of faculty members and have been included in the current study to serve as immutable characteristics of institutions and faculty. This approach is grounded in the literature about attitude formation where researchers Cunningham, Zelazo, Packer, & van Bavel [31] proposed a theory that even "current evaluations are constructed from relatively stable attitude representation". This combines the position that attitude formation (in this case, judgements made about benefits and barriers to undergraduate research mentoring) is a combined process involving long-term memory and current state assessment of a phenomenon. The main assumption in this study is that the benefits gained by mentoring an additional student are weighed against the costs (time, expense) of doing so when faculty consider whether or not to mentor an undergraduate student. This is in the context of SET where the benefits are defined as in the subscales, personal, professional and tenure and promotion. In contrast, the costs of the SET model involve the subscales institutional value and student-based. We theorize that

these benefits and barriers (costs) will mediate the relationship between research funding and number of students mentored due to the complex considerations each faculty mentors makes when deciding whether to advise another student. When provided with limited or no resources for accomplishing research tasks, faculty members will mentor fewer students [32]. When provided additional research support or other tangible benefits for mentoring undergraduates, faculty members will mentor more students. Inclusion of this variable also allows for analysis of factors that measure research group attributes that go beyond being resource rich and being able to support many undergraduate students in research. The weighting of costs and benefits guides the process for faculty members. Additionally, given the empirical evidence provided in the Monarrez et al. [10] study, we can now assess how these perceived benefits and barriers play out in practice by utilizing the SET framework.

To better understand why faculty members mentor undergraduate students in URE settings, we change the focus from past studies which gauged faculty perceptions about the motivations, benefits, and barriers to engaging in undergraduate research mentoring and instead use an empirical approach to understand which factors practically influence the number of students mentored in UREs and can mediate the effect of research funding. This is a complex and multi-faceted problem. Hence, we use a multi-layered process to understand the variety of factors and how interactions between these factors may impact the number of students mentored. Though the focus is on understanding factors associated with the number of undergraduate students mentored, we also model the number of graduate students mentored by the same cohort of biomedical research faculty included in the study. The inclusion of graduate students in a separate, but parallel, analysis will create a model for comparison. Moreover, use of this comparison model will illuminate any differences in motivation and beliefs of faculty when deciding whether to mentor an undergraduate versus a graduate student. This is also an appropriate approach since there may be a "preference" among some biomedical research faculty for mentoring graduate vs undergraduate students. Any potential preference for mentoring graduate students is likely due to the perception that they have more specialized knowledge and skills related to research, are more committed to research, and have more time to dedicate to research compared to undergraduate students. However, this ignores the evidence that many faculty value mentoring undergraduate students [10] and may prefer to mentor undergraduates even if they hold these same perceptions. There doesn't appear to be any documentation of this in the literature, and this study would be the first to make such a comparison and analyze any such preference in this regard.

In summary, this research study will provide valuable evidence about the most salient factors influencing the number of students mentored, with a focus on how these factors play out at a high research activity Hispanic-Serving Institution (HSI). In particular, we will answer to what extent the role of research funding has on increasing mentoring of undergraduate students and whether it is somehow impacted by the beliefs and barriers to engaging in research mentoring. Institutions and URE programs can use the results of this study to better address these factors with their faculty to ensure that every faculty member who wishes to mentor undergraduate researchers has the opportunity to do so, thereby providing broader access to students, particularly those in biomedical research training programs. In the following statement of the research hypotheses, make note that "students" refer both to undergraduate and graduate students, and the research questions will apply to both cohorts.

Thus, the primary research question of this study is:

*RQ 1: Do perceptions of benefits and barriers to undergraduate mentoring mediate the influence of research funding on the number of students (undergraduate and graduate) mentored by a biomedical research faculty?*

The secondary research question is:

*RQ 2: Are faculty characteristics associated with the number of students (undergraduate and graduate) mentored when controlling for faculty perceptions of barriers and benefits and baseline research funding?*

## Methodology

### Setting

Faculty included in this study are biomedical researchers at a large research-one HSI. We use the term biomedical researcher in the broadest sense, given that the faculty included conduct research that is considered biomedical, and may be appointed in biology, chemistry, physics mathematics, computer science, sociology, psychology, biomedical engineering or various health science disciplines. The quality enhancement plan of the university emphasizes experiential learning, and a historical strong focus of the institution is on research experiences for undergraduates. The faculty members included in this study are engaged in research mentoring of undergraduate or graduate students and may or may not receive funding from a structured research program. Almost all faculty members included in this study, however, regularly apply for grant support from state and federal agencies. Consent was informed for all research participants at each stage of the data collection and consent was obtained using written consent of each respondent. No minors were included in the data collection or study.

### Measures

The surveys used in this study were administered to biomedical research faculty members in the fall semesters of 2015, 2017, 2019, 2021, and 2023. The inclusion criteria for the study was faculty members that were or had previously engaged in undergraduate research mentoring. The window for participation in undergraduate mentoring included the years 2011 to 2023. This ensures that we are only including faculty who have some interest in undergraduate research mentoring and have structures in place to support the practice, even if they do not mentor undergraduates every year. Some faculty members also mentor graduate students and this information was recorded, though not as an inclusion criterion. Faculty members completed at least two of the years spanning 2015 to 2023. For these faculty with two or more instances of completing the faculty survey with at least one instance in 2015 or 2017, we retained the most recent number of mentored undergraduates and graduate students and computed the mean of the benefits and barriers to undergraduate research mentoring measures that preceded the year corresponding to the reported number of mentees. This preserves any temporal relationship between the mediator (benefits and barriers) and the responses (numbers of mentees) but also smooths the benefits and barriers measures. The number of undergraduate and graduate student mentees per faculty mentor had levels 0, 1–4, 5–9, and 10+.

The surveys included information about faculty characteristics, such as gender, race/ethnicity, college affiliation, rank, number of undergraduate and graduate mentees and duration of mentoring relationships. The surveys also asked for information about the students mentored, such as their classification and the inclusion of undergraduate and graduate students in presentations and publications. A set of questions also gathered faculty members' responses on research-based barriers and benefits of research mentoring. Faculty respondents rated their level of agreement using a five-point Likert scale (from strongly disagree to strongly agree). The details of the identification and validation of these self-reported barriers and benefits are available in Monarrez et al. [10]. However, we note that we identified three benefits and two

barriers to engaging undergraduates in research among research active biomedical faculty members. One of the perceived benefits was tenure and promotion (TP), where faculty members indicated they received credit for or merit for undergraduate research mentoring. Another benefit to undergraduate mentoring was personal benefits (P), where faculty members indicated they enjoyed the one-on-one mentoring process. The third benefit was professional and research (PR), where faculty members indicated that having undergraduate mentees helped advance and accelerate their research. One of the two barriers to undergraduate mentoring was institutional value (IV), which indicates faculty members' perception that the institution, department, or faculty peers do not value undergraduate research. The last barrier, student-based (SB), groups expressions of how undergraduates cannot or are not sufficiently well prepared to contribute to research outcomes. Monarrez [10] found that gender, rank, and college affiliation affected the level of agreement or disagreement with these five subscales regarding the benefits and barriers of undergraduate research mentoring.

Some responses had the scales reversed so that all high levels of the Likert scale indicate a higher level of agreement or support for undergraduate research mentoring. For example, if a question stem was negative, such as "My institution does not value undergraduate mentoring" with 1 indicating strongly disagree and 6 indicating strongly agree, then these scores were reversed so that a 1 indicates lack of perceived institutional value and 6 indicates a strong sense of institutional value. Other variable transformations were made to improve the analysis. All scales used in the survey were assessed for construct validity using factor models and data mining techniques [33, 34] and found to be unidimensional with high degrees of reliability. Upon reviewing the scale response levels, it was evident that the faculty could be easily grouped into those with high levels and those with low levels of the attribute regarding benefits and barriers to undergraduate mentoring. This simplifies the subsequent analysis since the models will not indicate a "one unit change" in the benefit and barrier attribute, which can be difficult to interpret in this context. Instead, the benefits and barriers can be more directly interpreted as those with high vs low levels of that belief. Following the validation of the scale data, appropriate items were combined, resulting in a final set of 13 variables from the original set of 91 variables.

To control for the amount of external funding of faculty members and number of research mentees, we obtained data from the research office on campus and created the following categories for levels of funding: $0, less than 20K, 20K-50K, 50K-150K, 150K-500K, 500K+. The funding data was collected for the same date of the data collection on faculty perceptions about mentoring undergraduates in research. This is used as a control and to isolate the effect of other factors in the model, but it is not a focus of analysis. A total of 1459 faculty (200 in 2015, 251 in 2017, 320 in 2019, 335 in 2021, and 353 in 2023) were surveyed with only 429 responding. Note that many of the faculty were surveyed each year included and thus the figure 1459 is not a unique number of faculty, but the total number surveyed over the years. The faculty surveyed were those in disciplines broadly classified as biomedical according to NIH classifications [35]. Out of the 419 faculty responses collected with multiple repeated measures, a total of 151 were unique faculty member responses that had started answering the survey in either 2015 or 2017 (35 in 2015, 41 in 2017, 37 in 2019, 19 in 2021, and 10 in 2023) and the records retained for analysis.

## Data validation and screening

After the data from the surveys were merged, we screened for any data entry errors and missing values. Of the 151 unique faculty responses there were no missing values among the set of variables to be used in the research study. This is a data cleaning and preparation step and

should be distinguished from the subsequent model-based imputation we integrate into the causal model outlined in the next section.

## Statistical analysis

The factors influencing the number of students mentored in research are assessed using causal models from a counterfactual perspective. In a counterfactual framework, it is necessary to hypothesize which factors are natural direct effects and which are natural indirect effects. In modeling, a natural direct effect is the conditional association between an exposure and outcome given the presence of a mediator. In contrast, a natural indirect effect combines the cumulative impact of the exposure and mediator on the outcome. Thus, when we speak of a mediating effect to funding levels of faculty, we are referring to the natural indirect effect of a factor that modifies the impact of funding on the number of students mentored. For example, in Moon & Moon [36], they provide an accessible example of direct and indirect effects: a caterpillar can eat a plant and directly impact the health of the plant (direct effect). A bird may eat the caterpillar and keep the caterpillar from harming the plant, thus indirectly impacting the plant (indirect effect). Differentiation of the direct and indirect effects assists in understanding complex systems, such as URE mentoring in university settings, by recognizing the variety of effects on our outcome of interest, number of students mentored. In observational data settings, the identifiability and exchangeability of the direct and indirect effects is difficult and can cause problems interpreting these models [37]. However, others have recently addressed these problems by defining so-called pure or natural direct and indirect effects [38]. These "natural" effects allow for a weaker set of assumptions made to ensure the identifiability for both types of effects even when there are confounding variables that may impact the exposure-outcome and/or mediator-outcome pairs [39]. This is a useful property since we have information from a very complex system, e.g. faculty member characteristics, their beliefs about mentoring students, and funding amounts. We believe all of these measures could be related to the outcome, number of students mentored. However, we do not believe that we have identified all possible confounding variables (or even can observe many of them) but still would like to distinguish among these effects with regards to their relative impact on the number of students mentored.

To define the model, we need to specify a mediator variable (M), an exposure variable (X), a response variable (Y), and control variables (C). Eqs 1 and 2 define the sources of heterogeneity in Y and M. In the equation, g is a cumulative logistic regression function and regression parameters are denoted $\alpha$ and $\beta$. This modeling approach assumes that C is independent of X, which will be checked in the modeling process. For this context, the mediation variables are the specific barriers and benefits expressed by faculty members in the Monarrez [10] study (M in Eqs 1 and 2). The hypothesis is that these effects will directly impact the number of students mentored in research (Y in Eq 2). Research funding is included as a potential natural direct and indirect effect (via benefits and barriers) on number of mentored students (X in Eqs 1 and 2). Finally, we set faculty characteristics as baseline control variables in the models (C in Eqs 1 and 2). This parameterization reflects our belief that research funding level and faculty characteristics have a direct effect on the number of students mentored. However, research funding may be mediated by faculty beliefs about undergraduate research mentoring. For example, faculty who value undergraduate mentoring may allocate more funds to this mentoring than faculty with low levels of perceived benefit. Fig 1 illustrates the proposed relationships hypothesized in the causal models. We note that this approach to modeling is a flexible approach and will accommodate almost any parametric distribution of the explanatory and response variables [40]. In this study, the number of mentees is an ordinal variable with groupings 0, 1–4, 5–9, 10+ and is thus treated using a generalized linear model with a cumulative

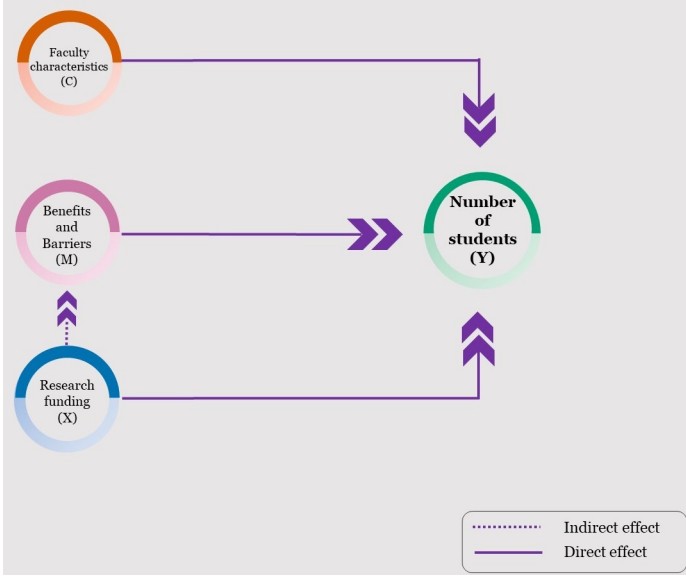

**Fig 1. Causal diagram describing natural direct and indirect effects on number of students mentored.**

logit link in the causal models (see Eq 2 below). In contrast, the mediator variable ranges from 1 to 10 and will be modeled using a linear model (see Eq 1).

$$g(P(M = m|X, C)) = \alpha_0 + \alpha_1 X + \alpha_2 C \tag{1}$$

$$g(P(Y = y|X, M, C)) = \beta_0 + \beta_1 X + \beta_2 M + \beta_3 C \tag{2}$$

The temporal effects of the model are well-identified, allowing a logical model with precursors to increasing the number of mentored students is research funding. However, perceptions of benefits and barriers to undergraduate research mentoring may affect the number of students ultimately mentored, which is also influenced by funding. This is the mediating pathway with funding being a direct effect, but with the total effect being mediated by the perceived benefits and barriers. Finally, faculty characteristics may also influence the benefits and barriers, which in turn also impact the number of students mentored in research.

**Chounterfactual framework.** Employing a counterfactual causal model allows for estimation of the unobserved effects, even variables that are lurking and could be confounding the exposure-outcome or mediator-outcome relationships. For example, if a faculty member receives $X = c$ level of funding, then $Y_i (X = c)$ is observed and all other potential values of $Y_i (X = c^*)$ are unobserved (where $c^*$ is another level of funding). An imputation model recovers the unavailable information so that the observed difference $Y_i(X = c) - Y_i(X = c^*)$ is estimable. In this case, this difference is the effect of funding on the number of undergraduate research mentees [40]. Employing an imputation approach for the counterfactuals, avoids relying on a specific model for the mediator distribution. Instead, the algorithm fits a working model for the outcome mean as defined by Vansteelandt et al. [41]. By setting $x^*$ (as opposed to $x$) equal to the observed exposure $X$, unobserved nested counterfactuals can be estimated using any suitable model for the outcome mean. In other words, as the potential intermediate outcome $M(x^*)$ is identical to the observed mediator $M$ within the subgroup with exposure $X = x^*$, $Y(x, M(x^*))$ is equivalent to $Y(x, M)$ for all individuals in that exposure group. This is a preferred approach since our mediator (the benefits and barriers) are not Gaussian and do not have a

characteristic structure to assume. Furthermore, in the call for the nested imputation, we can specify the formula for the multivariate GLM used for imputing the counterfactual outcomes. In particular, using the outcome model for the exposure group ($Y(x,M)$), the fitted values for $\hat{E}(Y|X = x, M, C)$ based on a polytomous logistic model for the funding levels using the mediator and control variates [41]. Details of the algorithm employed are in Table A1 of the S1 Appendix so as to not disrupt the flow of the manuscript. This is a clear and reasonable approach for multiple imputations of mixed data types (categorical and numerical) that is data-driven and reliable.

Subsequently, the latter can be estimated by using fitted values $\hat{E}(Y|X = x, M, C)$ based on an appropriate model for the outcome mean, referred to as the imputation model. In this model, exposure $X$ is set to $x$, and mediator $M$, along with baseline covariates $C$, is set to their observed values. This approach easily accommodates missing outcomes in the original dataset, as the corresponding nested counterfactuals can also be estimated in a similar manner.

We report model results using odds ratio estimates from the causal models. In particular, we will estimate the odds of going from one number (level) of student mentees to the next for those with one characteristic versus another. This is estimated using the slope parameter from the model, such as,

$$OR_{1,0}^{NDE} = \frac{odds\left(Y_{1_M} = 1|C\right)}{odds\left(Y_{0_M} = 1|C\right)} = e^{\beta}.$$

In the results, factor C, research funding, acts as a natural direct and indirect effect on the number of undergraduate students mentored. The faculty reported benefits and barrier mediators will be absorbed into this model via a term where the amount of funding includes these mediator effects. Thus, in the results, there will be two factors for funding, Funding (not mediated by benefits and barriers) and Funding (mediated by benefits and barriers), which indicate the natural direct effect of funding and natural indirect effect of funding (mediated by the benefits and barriers), respectively. In the model, the benefits and barriers are all coded, so that a higher level indicates a general "positive" outlook on undergraduate mentoring in these varied dimensions. This facilitates the global interpretation of the modeling results [42].

An exploratory analysis of the data revealed that the number of undergraduate and graduate research mentees has inappropriate variation in response to be adequately modeled using a traditional Poisson regression model. Specifically, for undergraduate and graduate research mentees, the ratios of the mean to the variance are 3.58 and 6.16, respectively. This indicates that the mean is inflated with respect to the variance and, thus, a quasi-Poisson modeling approach is appropriate to account for the lack of relative variation. Moreover, only four faculty mentors reported mentoring zero graduate students and only five reported mentoring 10 or more graduate students, thus we collapsed the 0 and 10 or more counts into the respective adjacent categories, resulting in two outcomes for number of graduate student research mentees: "0 to 4" and "5+". This will allow for a quasi-binomial approach for modeling, which is used only when modeling graduate student research mentees.

In the tabled results, factors associated with the number of research mentees are denoted by an asterisk based on the 95% confidence interval result. Whenever the confidence interval does not contain the value 1, the factor demonstrates association with the response. If the interval consists of values greater than 1, then the factor increases the probability of more research mentees and, if the interval includes values that are strictly less than 1, then the factor decreases the probability of more research mentees.

As an alternative mode of analysis, we performed an instrumental variable analysis using the benefits and barriers as separate (independent) factors influencing the relationship between funding and research mentoring of students. Keep in mind that in the counterfactual model, we could independently treat each component of the faculty reported benefits and barriers to have a specific and individualized impact on mediating the funding and research mentoring relationship. However, the instrumental variable approach is not as flexible in comparison and does not allow for this more nuanced modeling, unless we build an individual model for each of the five barriers and benefits (PR, R, P, IV, SB). This is an alternative view of the data used for the counterfactual modeling, but one that does not require additional imputation of responses for modeling the "what if" groups. Moreover, this approach also allows for the unmeasured variables that likely influence the research funding and mentoring culture to be incorporated into modeling. While we believe the counterfactual approach will provide additional power for detecting the influence of the "research climate" on the relationship between funding the research meeting due to its flexibility in the mediation structure and use of counterfactuals, we also regard this model as a valid alternative for investigating the negotiations made when faculty mentor students. These model results are provided in Tables A2 and A3 in S1 Appendix as a supplement to the model results presented.

All analysis is conducted in *R* (Version 4.1.2) [43] using the *medflex* package [44]. All results are reported using the model-based odds ratios for each factor and for univariate and multivariate settings making use of the benefits or barriers to mentoring students. P-values are provided with and without multiplicity corrections using a Šidàk correction to the p-value [45]. All model results are presented using the *gtsummary* package [46] in R and are produced directly from the estimated models. A power analysis was conducted and results indicate that the estimated model will only have 52% power for detecting true direct and indirect effects. See the study limitations section for a discussion about the implications and more information. Additionally, a sensitivity analysis is conducted on the final models presented for undergraduate and graduate mentees by using drop one variable analysis and bootstrap simulation. and a full stability analysis was performed after a final model was selected where bootstrap resamples of the model were used to assess the stability and reproducibility of the model results.

## Results

Table 1 presents descriptive statistics on the study participants and Table 2 presents information about their research and mentoring related characteristics. Note that the distribution across the variable levels is reasonable and that faculty represent a variety of sub-disciplines within the biomedical research sciences. When the data is categorical, the table presents the data with a frequency (n) and percentage (%). For example, for the barriers and benefits variables, we report just whether the faculty member held that belief with a binary variable, hence the frequency and percentage are reported. When the data are continuous, the median and interquartile range (IQR) are reported. Only half of faculty believe that undergraduate mentoring provides support for tenure and promotion, however more faculty believe in the other benefits of mentoring. Recall that these items were reversed when the item stem was negatively worded so that the presence of the attribute in this table is a "positive" view of mentoring undergraduates.

### Undergraduate student results

Using the multivariate adjusted model results provided in Table 3, there is evidence that funding increases the odds of more undergraduate mentees (OR(less than 20K) = 0.84, p-value<0.001; OR(20K-50K) = 0.654, p-value<0.001; OR(50K - 150K) = 1.32, p-value<0.001;

**Table 1. Faculty descriptive statistics.**

| Characteristic | N = 151[1] |
|---|---|
| **Race** | |
| White, Non-Hispanic | 56 (37%) |
| Hispanic | 22 (15%) |
| Other | 73 (48%) |
| **Faculty Rank** | |
| Assistant Professor | 29 (19%) |
| Associate Professor | 57 (38%) |
| Professor | 60 (40%) |
| Other | 5 (3.3%) |
| **Gender** | |
| Male | 55 (36%) |
| Female | 91 (60%) |
| Other | 5 (3.3%) |
| **Time at Work** | |
| % Research Time | 9 (7, 14) |
| % Administrative Time | 3.5 (2.0, 5.6) |
| **College Affiliation** | 15 (9.9%) |
| Engineering | 28 (19%) |
| Health Sciences | 36 (24%) |
| Liberal Arts | 72 (48%) |
| Science | |
| **Amount of funding** | |
| 0 | 41 (27%) |
| less than 20K | 6 (4.0%) |
| 20K-50K | 6 (4.0%) |
| 50K-150K | 21 (14%) |
| 150K-500K | 20 (13%) |
| more than 500K | 57 (38%) |

[1]n (%); Median (IQR)

OR(150K - 500K) = 1.16, p-value<0.001; OR(more than 500K) = 1.10, p-value<0.001). A very similar trend occurs with the univariate tests. However, at times, only the direct effect is significant and not the indirect mediated effect of funding levels with exceptions at moderate funding (50K-150K) and high level funding (more than 500K). Given that the benefits and barriers are explained by the underlying faculty characteristics, this is not surprising and implies that when these characteristics are controlled in the model, this effect disappears. Regarding faculty characteristics, Hispanic faculty are more likely than non-Hispanic faculty to mentor undergraduate students (OR(white) = 0.90, p-value<0.001) and males are also more likely to mentor more students than females (OR(male) = 0.96, p-value<0.001). This is true for both multivariate and univariate results. Being an associate or full professor decreases the odds of mentoring undergraduate students as well (OR(Assoc) = 0.99, p-value<0.001; OR(Full) = 0.96, p-value<0.001; OR(Other) = 0.88, p-value<0.001) and this association is true for both univariate and multivariate models. Finally, being in the college of health sciences (OR(HS) = 0.96, p-value<0.001) or college of science (OR(SCI) = 0.98, p-value<0.001) vs engineering increases the odds of mentoring more undergraduate students. There is potentially a slight increase for those who spend many hours performing administrative duties (for the multivariate model

**Table 2. Faculty research-related statistics.**

| Characteristic | N = 1511 |
|---|---|
| **Undergraduates Mentored** | |
| 0 | 5 (3.3%) |
| 1–4 | 60 (40%) |
| 5–9 | 48 (32%) |
| 10+ | 38 (25%) |
| **Graduates Mentored** | 38 (25%) |
| 0 | 5 (3.3%) |
| 1–4 | 76 (50%) |
| 5–9 | 55 (36%) |
| 10+ | 15 (9.9%) |
| **Benefits of UG Mentoring** | |
| TP | 4.00 (3.20, 5.33) |
| P | 2.50 (2.20, 3.25) |
| PR | 4.00 (3.25, 4.50) |
| **Barriers to UG Mentoring** | |
| SB | 11.67 (10.00, 12.40) |
| IV | 12.00 (10.75, 14.20) |

[1] n (%); Median (IQR)

results). However, this effect has only a marginal practical effect on mentoring practice due to the small effect size. The sensitivity analysis indicated that dropping a variable does not affect overall model interpretations, but only slightly modifies parameter estimators without changing overall interpretations. Outlying values did not influence model results significantly.

## Graduate student results

The results in Table 4, show a different pattern in mentoring graduate students as opposed to undergraduate students. In general, funding does not impact number of graduate students. However, keep in mind that these are all faculty active in research with either graduate or undergraduate mentees. This is also similarly observed in the univariate results. However, when mediated by the benefits and barriers, we see a small decrease in the number of graduate students mentored when faculty hold a "positive view" about undergraduate mentoring (OR(150K-500K) = 1.15, p-value<0.001; OR(more than 500K) = 1.12, p-value<0.001). Non-cis gender faculty tend to mentor graduate students at lower rates (OR (Other) = 0.77, p-value<0.001). Finally, being a full professor increases the odds of more graduate mentees (OR(full) = 1.09, p-value = 0.003) and College of Engineering faculty have higher odds of more graduate mentees than any other college. As with the undergraduate model, administrative time has a statistically significant (though practically small) positive effect on number of mentees for both univariate and multivariate results. Sensitivity analysis provides evidence of the stability of the model results under resampling via the bootstrap and by dropping single variables.

## Discussion

This study examines how faculty perceived benefits and barriers may mediate the effect of funding on the number of undergraduate research mentees. Because research teams and labs may sometimes reallocate funds between undergraduate and graduate research mentees, we

**Table 3. Univariate and multivariate fits of causal models for number of undergraduate students mentored.**

| | Univariate | | | Multivariate | | |
|---|---|---|---|---|---|---|
| Characteristic | OR | 95% CI[1] | p-value | OR | 95% CI[1] | p-value |
| **Funding (Not mediated by benefits and barriers)** | | | | | | |
| 0 | — | — | | — | — | |
| less than 20K | 0.84 | 0.81, 0.86 | <0.001[**,##] | 0.84 | 0.83, 0.84 | <0.001[**,##] |
| 20K-50K | 0.65 | 0.63, 0.67 | <0.001[**,##] | 0.65 | 0.65, 0.66 | <0.001[**,##] |
| 50K-150K | 1.32 | 1.29, 1.35 | <0.001[**,##] | 1.32 | 1.31, 1.33 | <0.001[**,##] |
| 150K-500K | 1.16 | 1.13, 1.19 | <0.001[**,##] | 1.16 | 1.15, 1.17 | <0.001[**,##] |
| more than 500K | 1.10 | 1.07, 1.13 | <0.001[**,##] | 1.10 | 1.09, 1.11 | <0.001[**,##] |
| **Funding (Mediated by benefits and barriers)** | | | | | | |
| 0 | — | — | | — | — | |
| less than 20K | 1.03 | 0.95, 1.12 | 0.500 | 1.02 | 1.00, 1.03 | 0.017 |
| 20K-50K | 1.03 | 0.95, 1.12 | 0.400 | 0.99 | 0.98, 1.01 | 0.400 |
| 50K-150K | 0.88 | 0.83, 0.93 | <0.001[**,##] | 0.95 | 0.94, 0.96 | <0.001[**,##] |
| 150K-500K | 1.00 | 0.95, 1.06 | 0.900 | 0.99 | 0.98, 1.00 | 0.040 |
| more than 500K | 0.92 | 0.88, 0.95 | <0.001[**,##] | 0.97 | 0.96, 0.98 | <0.001[**,##] |
| **Race/Ethnicity** | | | | | | |
| Hispanic | — | — | | — | — | |
| White | 0.90 | 0.86, 0.95 | <0.001[**,##] | 0.90 | 0.89, 0.90 | <0.001[**,##] |
| Other | 0.89 | 0.86, 0.92 | <0.001[**,##] | 0.87 | 0.86, 0.87 | <0.001[**,##] |
| **Sex** | | | | | | |
| Female | — | — | | — | — | |
| Male | 0.96 | 0.93, 0.99 | 0.012[*] | 0.96 | 0.95, 0.97 | <0.001[**,##] |
| Other | 0.98 | 0.89, 1.07 | 0.600 | 0.96 | 0.95, 0.98 | <0.001[**,##] |
| **Faculty Rank** | | | | | | |
| Assistant Professor | — | — | | — | — | |
| Associate Professor | 0.97 | 0.93, 1.01 | 0.120 | 0.99 | 0.98, 1.00 | 0.006 |
| Full Professor | 0.88 | 0.84, 0.92 | <0.001[**,##] | 0.96 | 0.95, 0.97 | <0.001[**,##] |
| Other | 0.82 | 0.74, 0.90 | <0.001[**,##] | 0.88 | 0.87, 0.90 | <0.001[**,##] |
| **Research Time** | 1.01 | 1.01, 1.01 | <0.001[**,##] | 1.01 | 1.01, 1.01 | <0.001[**,##] |
| **Administration Time** | 0.99 | 0.99, 1.0 | <0.001[**,##] | 0.99 | 0.99, 0.99 | <0.001[**,##] |
| **College Affiliation** | | | | | | |
| College of Engineering | — | — | | — | — | |
| College of Health Sciences | 1.10 | 1.03, 1.17 | 0.003[**] | 0.96 | 0.95, 0.98 | <0.001[**,##] |
| College of Liberal Arts | 1.05 | 0.99, 1.12 | 0.090 | 0.99 | 0.98, 1.00 | 0.064 |
| College of Science | 1.01 | 0.95, 1.07 | 0.800 | 0.98 | 0.96, 0.99 | <0.001[**,##] |

[*]Uncorrected p-value<0.05

[**]Uncorrected p-value<0.01

[#] Šidàk *Corrected p-value<0.05*

[##] Šidàk *Corrected p-value<0.01*

also model how faculty perceptions about undergraduate research mentoring indirectly affect the number of graduate student mentees. Flexible causal models are utilized to assess the degree of the natural direct and indirect effects of research funding on number of mentored students (undergraduate and graduate) with a mediating variable indicating level of perceived benefits and barriers.

**Table 4. Univariate and multivariate causal model fits for number of graduate students mentored.**

| Characteristic | Univariate | | | Multivariate | | |
|---|---|---|---|---|---|---|
| | OR | 95% CI[1] | p-value | OR | 95% CI[1] | p-value |
| **Funding (Not mediated by benefits and barriers)** | | | | | | |
| 0 | — | — | | — | — | |
| less than 20K | 1.00 | 0.94, 1.07 | >0.9 | 1.00 | 0.94, 1.06 | >0.9 |
| 20K-50K | 1.00 | 0.94, 1.07 | >0.9 | 1.00 | 0.94, 1.06 | >0.9 |
| 50K-150K | 1.00 | 0.94, 1.07 | >0.9 | 1.00 | 0.94, 1.06 | >0.9 |
| 150K-500K | 1.00 | 0.94, 1.07 | >0.9 | 1.00 | 0.94, 1.06 | >0.9 |
| more than 500K | 1.00 | 0.94, 1.07 | >0.9 | 1.00 | 0.94, 1.06 | >0.9 |
| **Funding (Mediated by benefits and barriers)** | | | | | | |
| 0 | — | — | | — | — | |
| less than 20K | 0.91 | 0.82, 1.00 | 0.065 | 0.88 | 0.78, 0.98 | 0.028 |
| 20K-50K | 0.91 | 0.82, 1.00 | 0.065 | 0.87 | 0.77, 0.97 | 0.014 |
| 50K-150K | 1.10 | 1.03, 1.16 | 0.003 | 1.12 | 1.05, 1.20 | 0.001 |
| 150K-500K | 1.13 | 1.06, 1.20 | <0.001 | 1.15 | 1.07, 1.23 | <0.001 |
| more than 500K | 1.09 | 1.04, 1.15 | <0.001 | 1.12 | 1.06, 1.19 | <0.001 |
| **Race/Ethnicity** | | | | | | |
| Hispanic | — | — | | — | — | |
| White | 0.98 | 0.93, 1.04 | 0.5 | 0.99 | 0.93, 1.05 | 0.7 |
| Other | 1.03 | 0.99, 1.07 | 0.15 | 1.01 | 0.96, 1.05 | 0.7 |
| **Sex** | | | | | | |
| Female | — | — | | — | — | |
| Male | 1.06 | 1.02, 1.10 | 0.005 | 0.97 | 0.93, 1.01 | 0.2 |
| Other | 0.90 | 0.80, 1.00 | 0.056 | 0.77 | 0.68, 0.87 | <0.001 |
| **Faculty Rank** | | | | | | |
| Assistant Professor | — | — | | — | — | |
| Associate Professor | 1.07 | 1.02, 1.13 | 0.012 | 1.05 | 0.99, 1.11 | 0.080 |
| Full Professor | 1.14 | 1.08, 1.20 | <0.001 | 1.09 | 1.03, 1.16 | 0.003 |
| Other | 0.94 | 0.83, 1.05 | 0.3 | 0.91 | 0.80, 1.03 | 0.14 |
| **Research Time** | 1.01 | 1.00, 1.01 | <0.001 | 1.00 | 1.00, 1.01 | 0.003 |
| **Administration Time** | 1.00 | 1.00, 1.01 | 0.040 | 1.00 | 1.00, 1.00 | >0.9 |
| **College Affiliation** | | | | | | |
| College of Engineering | — | — | | — | — | |
| College of Health Sciences | 0.87 | 0.81, 0.93 | <0.001 | 0.90 | 0.83, 0.97 | 0.006 |
| College of Liberal Arts | 0.85 | 0.80, 0.91 | <0.001 | 0.87 | 0.81, 0.93 | <0.001 |
| College of Science | 0.88 | 0.83, 0.93 | <0.001 | 0.83 | 0.78, 0.89 | <0.001 |

*Uncorrected p-value<0.05

**Uncorrected p-value<0.01

# Šidàk *Corrected p-value<0.05*

## Šidàk *Corrected p-value<0.01*

Overall, we see that funding levels affect the number of students (undergraduate and graduate) mentored and that the self-reported perceptions about undergraduate mentoring do not strongly affect the number of mentees at either the undergraduate or graduate level. Thus, our primary research question is fully answered with no to little evidence of faculty perception on actual number of research mentees. However, for both models, we do identify faculty characteristics, related to the benefits and barriers that affect the number of research mentees. This

explains why there is one mediated association for the undergraduate mentees in the univariate results that is controlled for in the multivariate model. Similarly, the graduate student model does indicate that more students are mentored with larger amounts of funding. This is a practical issue and could be due to the higher salaries and time investment required for mentoring graduate students. Otherwise, both models (for undergraduate and graduate students) indicate that faculty-level characteristics do affect the number of mentored students in the following ways. In general, men have higher odds of mentoring undergraduates and slightly lower odds of mentoring graduate students than female mentors. This is an interesting effect to observe considering that female faculty had more favorable beliefs about the benefits of undergraduate mentoring than male faculty [9]. The college of the faculty mentor also influences the number, with engineering faculty clearly favoring graduate student mentees and science and liberal arts favoring undergraduate mentees. This result echoes other findings where those in life sciences favored mentoring undergraduate mentees [21]. The variable indicating level of administrative time has a positive effect for both undergraduate and graduate mentees (the item asks proportion of time spent on administrative tasks). There may be an interpretation effect for this variable whereby faculty understand mentoring as part of the administration of their lab (hence positively associated with mentoring students).

This study sheds light on the importance of funding support in encouraging mentoring of undergraduate students among biomedical faculty. Indeed, previous research has established that having funding can positively predict the faculty's inclusion of undergraduates in research [22, 28, 47, 48]. The evidence provided in this study goes one step further to suggest that even small amounts of funding can have consistent and uniform positive effects by enabling biomedical research faculty to engage in and support undergraduate research mentees in projects. In contrast, there is little effect due to perceived barriers and benefits among these faculty for either undergraduate or graduate student mentees. Thus, the models suggest that small amounts of monetary support for research teams or labs is all that most faculty members need to start and continue mentoring undergraduate students. Since the number of undergraduate research mentees was unaffected by funding amount once the support exceeded 0, it is evident that any amount of funding increases the number of undergraduate research mentees among biomedical research faculty.

Regarding the SET framework motivating this study, this implies that when faculty navigate the pros and cons of research mentoring undergraduate students, that the primary driver in the decision-making process is ultimately funding. While we personally feel that improving faculty views on the benefits of undergraduate research mentoring is still a worthy cause, it probably has limited utility for increasing the numbers of research mentees. Perhaps additional efforts could be put into the funding and mentor training of research mentors so that they are enabled to fund undergraduate mentees and are prepared to be an effective mentor. This could redirect efforts of institutions to provide small amounts of funding to faculty who are willing to mentor undergraduates, but lack adequate funding.

Recall that this study involved biomedical research faculty from a diverse set of fields, including biology, physics, psychology, sociology, mathematics, computer science, biomedical engineering and chemistry. Though some fields require large amounts of money to support a lab (e.g., biology, chemistry, biomedical engineering) this study makes clear that even among these faculty, a small amount of funding has impact and encourages faculty to begin mentoring undergraduate students. Provided the importance of mentoring undergraduate researchers, this is an encouraging and positive result that should motivate institutions to develop grant-writing mentoring programs to improve faculty members' funding success and improve their capacity and efficiency of proposal submissions.

## Study limitations

We note that this study took place in a large R1 HSI institution with already strong biomedical research labs. The study results may not generalize to institutions in a very different context. Moreover, the study participants provided voluntary responses to the survey and so responses may be subject to volunteer bias. The original data used in the analysis was limited to only 85 faculty members and power analysis suggests the model is underpowered and may not reveal all true effects present. Using the procedure outlined in [49] we estimate an empirical power of only 52% using the observed effect and dependency structure and sample size. This implies there may be additional effects that do not appear to be significant due to lack of power. Larger scale studies may reveal additional factors that affect the number of students mentored in research.

## Supporting information

**S1 Appendix.**
(DOCX)

## Author Contributions

**Conceptualization:** Angelica Monarrez, Lourdes Echegoyen, Danielle Morales, Amy Wagler.

**Data curation:** Maria Aleida Ramirez, Diego Seira, Amy Wagler.

**Formal analysis:** Maria Aleida Ramirez, Diego Seira, Amy Wagler.

**Funding acquisition:** Amy Wagler.

**Investigation:** Lourdes Echegoyen, Danielle Morales, Amy Wagler.

**Methodology:** Amy Wagler.

**Project administration:** Angelica Monarrez, Amy Wagler.

**Resources:** Amy Wagler.

**Software:** Amy Wagler.

**Supervision:** Amy Wagler.

**Visualization:** Maria Aleida Ramirez, Amy Wagler.

**Writing – original draft:** Angelica Monarrez, Danielle Morales, Amy Wagler.

**Writing – review & editing:** Angelica Monarrez, Lourdes Echegoyen, Amy Wagler.

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
