## [Decision Letter · Decision Letter 0]

12 Sep 2023

PONE-D-23-20451Factors and mediators impacting the number of undergraduate research mentees at a research-intensive Hispanic-serving institutionPLOS ONE

Dear Dr. Wagler,

Thank you for submitting your manuscript to PLOS ONE. After careful consideration, we feel that it has merit but does not fully meet PLOS ONE’s publication criteria as it currently stands. Therefore, we invite you to submit a revised version of the manuscript that addresses the points raised during the review process.

We look forward to receiving your revised manuscript.

Kind regards,

Muhammad Khalid Bashir, PhD

Academic Editor

PLOS ONE

Journal Requirements:

**Additional Editor Comments:**

Authors need to address major concerns raised by Reviewer 1. Furthermore, they need to explain and justify sampling procedure and missing data issues as well. 

Reviewers' comments:

Reviewer's Responses to Questions

**Comments to the Author**

1. Is the manuscript technically sound, and do the data support the conclusions?

Reviewer #1: Partly

Reviewer #2: Yes

2. Has the statistical analysis been performed appropriately and rigorously? 

Reviewer #1: No

Reviewer #2: Yes

3. Have the authors made all data underlying the findings in their manuscript fully available?

Reviewer #1: No

Reviewer #2: Yes

4. Is the manuscript presented in an intelligible fashion and written in standard English?

Reviewer #1: Yes

Reviewer #2: Yes

5. Review Comments to the Author

Reviewer #1: Thank you for the opportunity to read the study titled “Factors and mediators impacting the number of undergraduate research mentees at a research-intensive Hispanic-serving institution.” The goal of the study was to deepen our understanding of how funding, faculty attitudes on benefits/barriers to undergraduate/graduate mentoring, demographic characteristics influence the number of students mentored. The study was helpfully situated within Social Exchange Theory and conducted at a large R1 HIS institution. Overall, I found the goal and scope of the study to be quite compelling and suspect it would be of interest to a wide audience. I found the study to be well written and appreciated the attention paid to statistical modeling of results. However, I had a number of concerns that tempered my enthusiasm for the findings. I’ll provide a review of those issues below.

Major issues:

The Methods (lines 210-216), make clear that the study authors used an inclusion criteria of “had an undergraduate mentee” for this study. It’s not clear if faculty without undergraduate mentees were surveyed or not. But this sample selection restriction systematically excluded faculty with zero undergraduate mentees. Given the study aims to estimate the causal effect of funding (X) on perceived benefits (M), and number of URE mentees in a biomedical lab (Y). Playing devil's advocate, let's imagine that the causal relationships are zero. By only surveying (or including) faculty that currently have a URE mentee, you've controlled for "currently has a URE mentee." To have a URE mentee, you need funding (X), to perceive some benefit (M), have some number of mentees (Y), or all three. Therefore, currently having a URE mentee is a collider in the data generating process and controlling for it may create selection bias and thereby induce correlations out of nothing - or in the present case may distort causal relationships that are present.

An additional concern relates to omission of key information on the faculty surveyed. How many biomedical faculty are at the university (i.e., population). What was the response rate does N = 154 represent? Are there any differences between the analytic sample (n = 85) and the total population, in terms of demographics, disciplines, etc?

A further concern relates to the high amount of missing data reported by those that did respond to the survey. The description / characterization of the missing data patterns on a variable-by-variable basis would be helpful. Furthermore, reporting the MCAR test is needed. Finally the description of the imputation model is insufficiently incomplete. If imputation is used, multiple imputation is a preferred approach over single imputation models - please describe the approach in detail and report key characteristics like iteration, burn-in rate, and post-imputation diagnostics to assure readers that the imputation model is justified and reasonably accurate. See Enders or Graham for a useful descriptions.

Enders, C. K. (2010). Applied missing data analysis. Guilford Press.

Graham, J. W. (2012). Missing data: Analysis and design. Springer Science & Business Media. https://doi.org/10.1007/978-1-4614-4018-5

Finally, the statistical and causal logic of the two main equations (lines 320-326) is somewhat concerning. Specifically, the temporal effects between the mediator (faculty perceptions) and outcome (number of students) is not temporally well-identified. The two are contemporaneous at best, but are more likely backward in this data set. That is the current number of mentees is based on a decision that occurred in the past (number of mentees represents a choice made in the past), while the perceptions are current (that is determined at the time of the survey). Therefore, the causal direction between these two variables is not identifiable in this data set (unless there is some additional data not discussed here in which the number of mentees is assessed at some point in the future).

Minor concerns:

The Theoretical Framework section focuses on the potential explanatory relevance of SET – and I agree that this could be a useful framework. However, little in the introductory narrative or in the methods links the constructs of SET (as typically measured) to what was measured in the current study. I encourage the authors to spend a bit more time connecting the theory to the nomological network under SET and how that was operationalized in terms of key constructs in this study.

Line 314-315 – is there some reason for treating the number of mentees as ordinal, when this is a count variable? Was this how the question was asked (i.e., ordinal groupings)?

Line 325-326 – the authors believe that faculty characteristics influence perceptions of benefits and barriers, but this isn’t shown in the FIG 1 DAG.

Line 350 – the distributional issues / characteristics of the ordinal outcome may be due to the selection criteria omitting faculty with no current undergraduate mentees.

Reviewer #2: The concept is well perceived but it lacks a clear research gap identification. Methods and analysis is appropriate. Manuscript is written in a scholarly style. presentation of results and discussion is really good.

6. PLOS authors have the option to publish the peer review history of their article (what does this mean?). If published, this will include your full peer review and any attached files.

Reviewer #1: No

Reviewer #2: No

---

## [Author Response · Author response to Decision Letter 0]

28 Mar 2024

Reviewer #1: 

Thank you for the opportunity to read the study titled “Factors and mediators impacting the number of undergraduate research mentees at a research-intensive Hispanic-serving institution.” The goal of the study was to deepen our understanding of how funding, faculty attitudes on benefits/barriers to undergraduate/graduate mentoring, demographic characteristics influence the number of students mentored. The study was helpfully situated within Social Exchange Theory and conducted at a large R1 HIS institution. Overall, I found the goal and scope of the study to be quite compelling and suspect it would be of interest to a wide audience. I found the study to be well written and appreciated the attention paid to statistical modeling of results. However, I had a number of concerns that tempered my enthusiasm for the findings. I’ll provide a review of those issues below.

Major issues:

Major Issue 1: The Methods (lines 210-216), make clear that the study authors used an inclusion criteria of “had an undergraduate mentee” for this study. It’s not clear if faculty without undergraduate mentees were surveyed or not. But this sample selection restriction systematically excluded faculty with zero undergraduate mentees. Given the study aims to estimate the causal effect of funding (X) on perceived benefits (M), and number of URE mentees in a biomedical lab (Y). Playing devil's advocate, let's imagine that the causal relationships are zero. By only surveying (or including) faculty that currently have a URE mentee, you've controlled for "currently has a URE mentee." To have a URE mentee, you need funding (X), to perceive some benefit (M), have some number of mentees (Y), or all three. Therefore, currently having a URE mentee is a collider in the data generating process and controlling for it may create selection bias and thereby induce correlations out of nothing - or in the present case may distort causal relationships that are present.

Major Issue 2: An additional concern relates to omission of key information on the faculty surveyed. How many biomedical faculty are at the university (i.e., population). What was the response rate does N = 154 represent? Are there any differences between the analytic sample (n = 85) and the total population, in terms of demographics, disciplines, etc?

Major Issue 3: A further concern relates to the high amount of missing data reported by those that did respond to the survey. The description / characterization of the missing data patterns on a variable-by-variable basis would be helpful. Furthermore, reporting the MCAR test is needed. Finally the description of the imputation model is insufficiently incomplete. If imputation is used, multiple imputation is a preferred approach over single imputation models - please describe the approach in detail and report key characteristics like iteration, burn-in rate, and post-imputation diagnostics to assure readers that the imputation model is justified and reasonably accurate. See Enders or Graham for a useful descriptions.

Enders, C. K. (2010). Applied missing data analysis. Guilford Press.

Graham, J. W. (2012). Missing data: Analysis and design. Springer Science & Business Media. https://doi.org/10.1007/978-1-4614-4018-5

Major Issue 4: Finally, the statistical and causal logic of the two main equations (lines 320-326) is somewhat concerning. Specifically, the temporal effects between the mediator (faculty perceptions) and outcome (number of students) is not temporally well-identified. The two are contemporaneous at best, but are more likely backward in this data set. That is the current number of mentees is based on a decision that occurred in the past (number of mentees represents a choice made in the past), while the perceptions are current (that is determined at the time of the survey). Therefore, the causal direction between these two variables is not identifiable in this data set (unless there is some additional data not discussed here in which the number of mentees is assessed at some point in the future).

Minor concerns:

The Theoretical Framework section focuses on the potential explanatory relevance of SET – and I agree that this could be a useful framework. However, little in the introductory narrative or in the methods links the constructs of SET (as typically measured) to what was measured in the current study. I encourage the authors to spend a bit more time connecting the theory to the nomological network under SET and how that was operationalized in terms of key constructs in this study.

Line 314-315 – is there some reason for treating the number of mentees as ordinal, when this is a count variable? Was this how the question was asked (i.e., ordinal groupings)?

Line 325-326 – the authors believe that faculty characteristics influence perceptions of benefits and barriers, but this isn’t shown in the FIG 1 DAG.

Line 350 – the distributional issues / characteristics of the ordinal outcome may be due to the selection criteria omitting faculty with no current undergraduate mentees. Response to Reviewer #1:

First of all, thank you for your thoughtful and constructive review of the manuscript. We appreciate your attention to our contribution. We appreciate that you were interested in and focused on refining our approach for this paper.

We note that we took considerable time making these revisions because we put in a request for more data from the evaluation team of the project. We were unaware that they had continued this survey and so were pleasantly surprised that some of the issues you identify were resolvable using the additional data provided by the evaluation team. Thank you again for your thoughtful and constructive comments!

Major issue 1: Thank you. This is an excellent point and prompted me to think about this issue for quite some time! In response, I have to say we were not careful with wording since there were faculty included who had been mentoring undergraduates but had since quit (for unknown reasons) as well as faculty currently engaged. When we wrote the original phrase we should have clarified that the pool was faculty who were or had been engaged in UG mentoring. I verified in the data as well. Now our control is “has or had a URE mentee”. We note too that at our institution, there are many faculty who do not have funding but still have a URE mentee. Although this is somewhat discipline specific. The wording of the section is changed to the following:

“The inclusion criteria for the survey were that faculty members were or had been currently engaged in undergraduate research mentoring. The window for participation in undergraduate mentoring included the years 2011 to 2017. This ensures that we are only including faculty that have some interest in undergraduate research mentoring and have structures in place to support the practice.”

Major Issue 2: Thank you for recognizing this omission. In the current draft, there is an added table that characterizes the sample of faculty in the study and the difference in the full population vs response and those we included. We hope this satisfies this issue sufficiently.

Major Issue 3: Thank you for your very helpful comments here. Though we ran an imputation model, we did not include enough information to summarize. In the original analysis only 4 observations had missing values for college affiliation and gender. We did not perform the standard MCAR test (Little’s) since it is not valid for categorical features and shouldn’t be reported. However, we used visual inspections and, given the magnitude performed a multiple imputation. However, we added additional data for this revision (see Major Issue 4) and now the missing data cases (all 4) are resolved and we have no missing data. All of these additions are included as tracked changes in the manuscript. 

However, we did include much more explanation about the imputations used for the counterfactual modeling. We wrongly omitted that in the manuscript, but it is now included. 

Major Issue 4: Thank you. This is a good point about the lack of temporal justification. However, when we set up the models, we regarded the benefits and barriers to be a relatively static or immutable mediator that wasn’t subject to these concerns. Our previous analysis provided evidence that these are long-standing (mostly invariant over time) dispositions among researchers that determine their openness and receptivity to mentoring UG students. This was our working assumption at the time of modeling. In fact, this is a common assumption and one that is supported in the literature and now cited in the paper.

However, we did take action to address your very valid concerns and found additional data (that I did not know existed!) that allowed us to include the previous year’s survey results about benefit and barriers so the temporal issue is no longer a problem. However, we had a lengthy discussion in the group of authors and decided to confirm the modeling approach using counterfactuals using a DAG and a instrumental variable model. 

The alternative analytical approaches relax and/or check the assumption of conditional independence of the mediator and response given the treatment. The DAG verified our approach when constructed using the benefits and barriers separately and showed a direct and indirect pathway of funding on UG and G research mentoring. Moreover, the IV approach was beneficial to look at since it allows the mediator (benefits and barriers) to be associated with the response (# students mentored) in a way that permits a different perspective of the current data. Upon reflection, we now feel the counterfactual modeling approach is, in fact, a more valid option with the data changes you suggested. However, we are glad we investigated the IV approach since a more general disposition of faculty towards research and students at large may also play a role and reflect the scores we obtained on the benefits and barriers scale. Thus, we feel confident that the newly reported model with the additional data included does in fact reflect current evidence about the true relationship between faculty funding and UG student research mentoring. All adjustments to the description of the statistical analysis performed are included as tracked changes in the manuscript. 

We included more description of how the SET framework ties to the subscales of the survey used. Thank you for pointing this out and please let us know if you would like additional details. 

Good point. I too would prefer to do count modeling, but the survey had mentors give the responses as reported in the paper, so it is not possible to use counts.

Faculty characteristics are at the top of the figure with an arrow directed to number of mentees.

There were faculty with no UG mentees since we surveyed all biomedical research faculty. The phrase leading you to believe this is corrected in the manuscript.

Reviewer #2: The concept is well perceived but it lacks a clear research gap identification. Methods and analysis is appropriate. Manuscript is written in a scholarly style. presentation of results and discussion is really good. Thank you for your review and feedback. We have added to the section showing the need for insight into this relationship between funding, faculty/mentor perspectives on UG mentoring and number of students mentored.

---

## [Decision Letter · Decision Letter 1]

11 Jun 2024

Factors and mediators impacting the number of undergraduate research mentees at a research-intensive Hispanic-serving institution

PONE-D-23-20451R1

Dear Dr. Wagler,

We’re pleased to inform you that your manuscript has been judged scientifically suitable for publication and will be formally accepted for publication once it meets all outstanding technical requirements.

Kind regards,

Muhammad Khalid Bashir, PhD

Academic Editor

PLOS ONE

Additional Editor Comments (optional):

Reviewers' comments:

Reviewer's Responses to Questions

**Comments to the Author**

1. If the authors have adequately addressed your comments raised in a previous round of review and you feel that this manuscript is now acceptable for publication, you may indicate that here to bypass the “Comments to the Author” section, enter your conflict of interest statement in the “Confidential to Editor” section, and submit your "Accept" recommendation.

Reviewer #1: All comments have been addressed

2. Is the manuscript technically sound, and do the data support the conclusions?

Reviewer #1: Yes

3. Has the statistical analysis been performed appropriately and rigorously? 

Reviewer #1: Yes

4. Have the authors made all data underlying the findings in their manuscript fully available?

Reviewer #1: Yes

5. Is the manuscript presented in an intelligible fashion and written in standard English?

Reviewer #1: Yes

6. Review Comments to the Author

Reviewer #1: Thank you for the opportunity to re-review the revised and much-improved manuscript titled “Factors and mediators impacting the number of undergraduate research mentees at a Research-intensive Hispanic-serving Institution.” I appreciate your consideration of my feedback and, overall, believe that the revisions have satisfied my prior concerns. I am particularly glad to see that the additional data you were able to gather and analyze helped alleviate the prior issue with temporal precedence. I have just one moderate and a couple of minor issues to bring to your attention.

Lingering issues

1. One moderate issue concerns your discussion of study limitations. While I agree with what you’ve mentioned already, reasonable potential confounds are missing from your discussion. For instance, your data are institutional in nature rather than direct observations or surveys from relevant stakeholders. It seems reasonable that psychological and motivational may also influence the relationship between the exposure, mediator, and outcome variables. There may be other factors listed in your DAG/data generating model that are relevant here too but missing from the administrative data. I suggest discussing reasonable confounds that could threaten your treatment direct/indirect effect.

2. Figure 1 appears to be missing, so please re-add this to the manuscript.

3. Many/most of your references are missing the DOI

7. PLOS authors have the option to publish the peer review history of their article (what does this mean?). If published, this will include your full peer review and any attached files.

Reviewer #1: No

---

## [Editor Report · Acceptance letter]

20 Jul 2024

PONE-D-23-20451R1 

PLOS ONE

Dear Dr. Wagler, 

I'm pleased to inform you that your manuscript has been deemed suitable for publication in PLOS ONE. Congratulations! Your manuscript is now being handed over to our production team.

Kind regards, 

on behalf of

Dr. Muhammad Khalid Bashir 

Academic Editor

PLOS ONE